# Mutual Information State Intrinsic Control

**Rui Zhao[1,2]\*, Yang Gao[3], Pieter Abbeel[4], Volker Tresp[1,2], Wei Xu[5]**
[1]Ludwig Maximilian University of Munich  [2]Siemens AG
[3]Tsinghua University  [4]University of California, Berkeley  [5]Horizon Robotics

## Abstract

Reinforcement learning has been shown to be highly successful at many challenging tasks. However, success heavily relies on well-shaped rewards. Intrinsically motivated RL attempts to remove this constraint by defining an intrinsic reward function. Motivated by the self-consciousness concept in psychology, we make a natural assumption that the agent knows what constitutes itself, and propose a new intrinsic objective that encourages the agent to have maximum control on the environment. We mathematically formalize this reward as the mutual information between the agent state and the surrounding state under the current agent policy. With this new intrinsic motivation, we are able to outperform previous methods, including being able to complete the pick-and-place task for the first time without using any task reward. A video showing experimental results is available at `https://youtu.be/AUCwc9RThpk`.

## 1 Introduction

Reinforcement learning (RL) allows an agent to learn meaningful skills by interacting with an environment and optimizing some reward function, provided by the environment. Although RL has achieved impressive achievements on various tasks (Silver et al., 2017; Mnih et al., 2015; Berner et al., 2019), it is very expensive to provide dense rewards for every task we want the robot to learn. Intrinsically motivated reinforcement learning encourages the agent to explore by providing an "internal motivation" instead, such as curiosity (Schmidhuber, 1991; Pathak et al., 2017; Burda et al., 2018), diversity (Gregor et al., 2016; Haarnoja et al., 2018; Eysenbach et al., 2019) and empowerment (Klyubin et al., 2005; Salge et al., 2014; Mohamed & Rezende, 2015). Those internal motivations can be computed on the fly when the agent is interacting with the environment, without any human engineered reward. We hope to extract useful "skills" from those internally motivated agents, which could later be used to solve downstream tasks, or simply augment the sparse reward with those intrinsic rewards to solve a given task faster.

Most of the previous works in RL model the environment as a Markov Decision Process (MDP). In an MDP, we use a single state vector to describe the current state of the whole environment, without explicitly distinguishing the agent itself from its surrounding. However, in the physical world, there is a clear boundary between an intelligent agent and its surrounding. The skin of any mammal is an example of such boundary. The separation of the agent and its surrounding also holds true for most of the man-made agents, such as any mechanical robot. This agent-surrounding separation has been studied for a long time in psychology under the concept of self-consciousness. Self-consciousness refers that a subject knows itself is the object of awareness (Smith, 2020), effectively treating the agent itself differently from everything else. Gallup (1970) has shown that self-consciousness widely exists in chimpanzees, dolphins, some elephants and human infants. To equally emphasize the agent and its surrounding, we name this separation as agent-surrounding separation in this paper. The widely adopted MDP formulation ignores the natural agent-surrounding separation, but simply stacks the agent state and its surrounding state together as a single state vector. Although this formulation is mathematically concise, we argue that it is over-simplistic, and as a result, it makes the learning harder.

With this agent-surrounding separation in mind, we are able to design a much more efficient intrinsically motivated RL algorithm. We propose a new intrinsic motivation by encouraging the agent to

---

\*Correspondence to: Rui Zhao {`zhaorui.in.germany@gmail.com`}.

perform actions such that the resulting agent state should have high Mutual Information (MI) with the surrounding state. Intuitively, the higher the MI, the more control the agent could have on its surrounding. We name the proposed method "MUtual information-based State Intrinsic Control", or MUSIC in short. With the proposed MUSIC method, we are able to learn many complex skills in an unsupervised manner, such as learning to pick up an object without any task reward. We can also augment a sparse reward with the dense MUSIC intrinsic reward, to accelerate the learning process.

Our contributions are three-fold. First, we propose a novel intrinsic motivation (MUSIC) that encourages the agent to have maximum control on its surrounding, based on the natural agent-surrounding separation assumption. Secondly, we propose scalable objectives that make the MUSIC intrinsic reward easy to optimize. Last but not least, we show MUSIC's superior performance, by comparing it with other competitive intrinsic rewards on multiple environments. Noticeably, with our method, for the first time the pick-and-place task can be solved without any task reward.

## 2 PRELIMINARIES

For environments, we consider four robotic tasks, including push, slide, pick-and-place, and navigation, as shown in Figure 2. The goal in the manipulation task is to move the target object to a desired position. For the navigation task, the goal is to navigate to a target ball. In the following, we define some terminologies.

### 2.1 AGENT STATE, SURROUNDING STATE, AND REINFORCEMENT LEARNING SETTINGS

In this paper, the **agent state** $s^a$ means literally the state variable of the agent. The **surrounding state** $s^s$ refers to the state variable that describes the surrounding of the agent, for example, the state variable of an object. For multi-goal environments, we use the same assumption as previous works (Andrychowicz et al., 2017; Plappert et al., 2018), which consider that the goals can be represented as states and we denote the goal variable as $g$. For example, in the manipulation task, a goal is a particular desired position of the object in the episode. These desired positions, i.e., goals, are sampled from the environment.

The **division** between the agent state and the surrounding state is naturally defined by the agent-surrounding separation concept introduced in Section 1. From a biology point of view, a human can naturally distinguish its own parts, like hands or legs from the environments. Analog to this, when we design a robotic system, we can easily know what is the agent state and what is its surrounding state. In this paper, we use upper letters, such as $S$, to denote random variables and the corresponding lower case letter, such as $s$, to represent the values of random variables.

We assume the world is fully observable, including a set of states $\mathcal{S}$, a set of actions $\mathcal{A}$, a distribution of initial states $p(s_0)$, transition probabilities $p(s_{t+1} \mid s_t, a_t)$, a reward function $r\colon \mathcal{S} \times \mathcal{A} \to \mathbb{R}$, and a discount factor $\gamma \in [0, 1]$. These components formulate a Markov Decision Process represented as a tuple, $(\mathcal{S}, \mathcal{A}, p, r, \gamma)$. We use $\tau$ to denote a trajectory, which contains a series of agent states and surrounding states. Its random variable is denoted as $\mathcal{T}$.

## 3 METHOD

We focus on agent learning to control its surrounding purely by using its observations and actions without supervision. Motivated by the idea that when an agent takes control of its surrounding, then there is a high MI between the agent state and the surrounding state, we formulate the problem of learning without external supervision as one of learning a policy $\pi_\theta(a_t \mid s_t)$ with parameters $\theta$ to maximize intrinsic MI rewards, $r = I(S^a; S^s)$. In this section, we formally describe our method, mutual information-based state intrinsic control (MUSIC).

### 3.1 MUTUAL INFORMATION REWARD FUNCTION

Our framework simultaneously learns a policy and an intrinsic reward function by maximizing the MI between the surrounding state and the agent state. Mathematically, the MI between the surround-

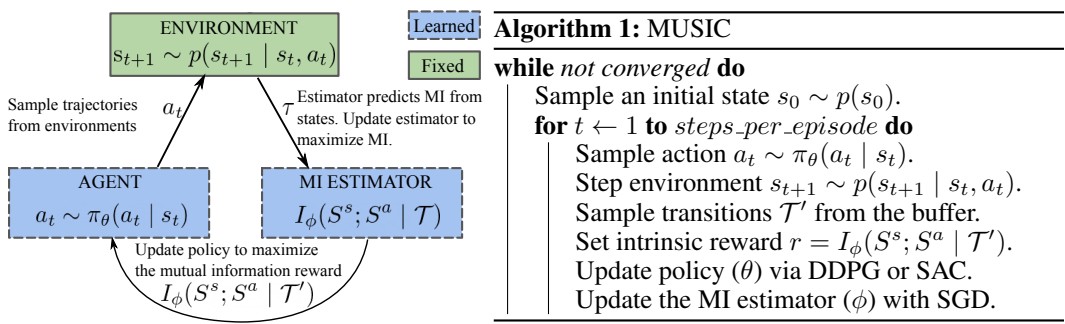

Figure 1: **MUSIC Algorithm**: We update the estimator to better predict the MI, and update the agent to control the surrounding state to have higher MI with the agent state.

ing state random variable $S^s$ and the agent state random variable $S^a$ is represented as follows:

$$I(S^s; S^a) = KL(\mathbb{P}_{S^s S^a} \| \mathbb{P}_{S^s} \otimes \mathbb{P}_{S^a}) \tag{1}$$

$$= \sup_{T:\Omega \to \mathbb{R}} \mathbb{E}_{\mathbb{P}_{S^s S^a}}[T] - \log(\mathbb{E}_{\mathbb{P}_{S^s} \otimes \mathbb{P}_{S^a}}[e^T]) \tag{2}$$

$$\geq \sup_{\phi \in \Phi} \mathbb{E}_{\mathbb{P}_{S^s S^a}}[T_\phi] - \log(\mathbb{E}_{\mathbb{P}_{S^s} \otimes \mathbb{P}_{S^a}}[e^{T_\phi}]) \coloneqq I_\Phi(S^s; S^a), \tag{3}$$

where $\mathbb{P}_{S^s S^a}$ is the joint probability distribution; $\mathbb{P}_{S^s} \otimes \mathbb{P}_{S^a}$ is the product of the marginal distributions $\mathbb{P}_{S^s}$ and $\mathbb{P}_{S^a}$; *KL* denotes the Kullback-Leibler (KL) divergence. MI is notoriously difficult to compute in real-world settings (Hjelm et al., 2019). Compared to the variational information maximizing-based approaches (Barber & Agakov, 2003; Alemi et al., 2016; Chalk et al., 2016; Kolchinsky et al., 2017), the recent MINE-based approaches have shown superior performance (Belghazi et al., 2018; Hjelm et al., 2019; Velickovic et al., 2019). Motivated by MINE (Belghazi et al., 2018), we use a lower bound to approximate the MI quantity $I(S^s; S^a)$. First, we rewrite Equation (1), the KL formulation of the MI objective, using the Donsker-Varadhan representation, to Equation (2) (Donsker & Varadhan, 1975). The input space $\Omega$ is a compact domain of $\mathbb{R}^d$, i.e., $\Omega \subset \mathbb{R}^d$, and the supremum is taken over all functions $T$ such that the two expectations are finite. Secondly, we lower bound the MI in the Donsker-Varadhan representation with the compression lemma in the PAC-Bayes literature and derive Equation (3) (Banerjee, 2006; Belghazi et al., 2018). The expectations in Equation (3) are estimated by using empirical samples from $\mathbb{P}_{S^s S^a}$ and $\mathbb{P}_{S^s} \otimes \mathbb{P}_{S^a}$. The statistics model $T_\phi$ is parameterized by a deep neural network with parameters $\phi \in \Phi$, whose inputs are the empirical samples.

## 3.2 EFFECTIVELY COMPUTING THE MUTUAL INFORMATION REWARD IN PRACTICE

**Lemma 1.** *There is a monotonically increasing relationship between* $I_\phi(S^s; S^a \mid \mathcal{T})$ *and* $\mathbb{E}_{\mathbb{P}_{\mathcal{T}'}}[I_\phi(S^s; S^a \mid \mathcal{T}')]$, *mathematically,*

$$I_\phi(S^s; S^a \mid \mathcal{T}) \ltimes \mathbb{E}_{\mathbb{P}_{\mathcal{T}'}}[I_\phi(S^s; S^a \mid \mathcal{T}')], \tag{4}$$

*where* $S^s$, $S^a$, *and* $\mathcal{T}$ *denote the surrounding state, the agent state, and the trajectory, respectively. The trajectory fractions are defined as the adjacent state pairs, namely* $\mathcal{T}' = \{S_t, S_{t+1}\}$. *The symbol* $\ltimes$ *denotes a monotonically increasing relationship between two variables and* $\phi$ *represents the parameter of the statistics model in MINE. Proof. See Appendix A.* □

We define the reward for each transition at a given time-step as the mutual information of the pair of adjacent states at that time-step, see Equation (4) Right-Hand Side (RHS). However, in practice, we find that it is not very efficient to train the MI estimator using state pairs. To counter this issue, we use all the states in the same trajectory in a batch to train the MI estimator, see Equation (4) Left-Hand Side (LHS), since more empirical samples help to reduce variance and therefore accelerate learning. In Lemma 1, we prove the monotonically increasing relationship between Equation (4) RHS and Equation (4) LHS.

In more detail, we divide the process of computing rewards into two phases, i.e., the training phase and the evaluation phase. In the training phase, we efficiently train the MI estimator with a large

batch of samples from the whole trajectory. For training the MI estimator network, we first randomly sample the trajectory $\tau$ from the replay buffer. Then, the states $s_t^a$ used for calculating the product of marginal distributions are sampled by shuffling the states $s_t^a$ from the joint distribution along the temporal axis $t$ within the trajectory. We use back-propagation to optimize the parameter ($\phi$) to maximize the MI lower bound, see Equation (4) LHS.

For evaluating the MI reward, we use a pair of transitions to calculate the transition reward, see Equation (4) RHS and Equation (5), instead of using the complete trajectory. Each time, to calculate the MI reward for the transition, the reward is calculated over a small fraction of the complete trajectory $\tau'$, namely $r = I_\phi(S^s; S^a \mid \mathcal{T}')$. The trajectory fraction, $\tau'$, is defined as adjacent state pairs, $\tau' = \{s_t, s_{t+1}\}$, and $\mathcal{T}'$ represents its corresponding random variable.

The derived Lemma 1 brings us two important benefits. First, it enables us to efficiently train the MI estimator using all the states in the same trajectory. And a large batch of empirical samples reduce the variance of the gradients. Secondly, it allows us to estimate the MI reward for each transition with only the relevant state pair. This way of estimating MI enables us to assign rewards more accurately at the transition level.

Based on Lemma 1, we calculate the transition reward as the MI of each trajectory fraction, namely

$$r_\phi(a_t, s_t) := I_\phi(S^s; S^a | \mathcal{T}') = 0.5\sum_{i=t}^{t+1}T_\phi(s_i^s, s_i^a) - \log(0.5\sum_{i=t}^{t+1}e^{T_\phi(s_i^s, \bar{s}_i^a)}), \qquad (5)$$

where $(s_i^s, s_i^a) \sim \mathbb{P}_{S^s S^a|\mathcal{T}'}$, $\bar{s}_i^a \sim \mathbb{P}_{S^a|\mathcal{T}'}$, and $\tau' = \{s_t, s_{t+1}\}$. In case that the estimated MI value is particularly small, we scale the reward with a hyper-parameter $\alpha$ and clip the reward between 0 and 1. MUSIC can be combined with any off-the-shelf reinforcement learning methods, such as deep deterministic policy gradient (DDPG) (Lillicrap et al., 2016) and soft actor-critic (SAC) (Haarnoja et al., 2018). We summarize the complete training algorithm in Algorithm 1 and in Figure 1.

**MUSIC Variants with Task Rewards:** The introduced MUSIC method is an unsupervised reinforcement learning approach, which is denoted as "MUSIC-u", where "-u" stands for unsupervised learning. We propose three ways of using MUSIC to accelerate learning. The first method is using the MUSIC-u pretrained policy as the parameter initialization and then fine-tuning the agent with the task rewards. We denote this variant as "MUSIC-f", where "-f" stands for fine-tuning. The second variant is to use the MI intrinsic reward to help the agent to explore more efficiently. Here, the MI reward and the task reward are added together. We name this method as "MUSIC-r", where "-r" stands for reward. The third approach is to use the MI quantity from MUSIC to prioritize trajectories for replay. The approach is similar to the TD-error-based prioritized experience replay (PER) (Schaul et al., 2016). The only difference is that we use the estimated MI instead of the TD-error as the priority for sampling. We name this method as "MUSIC-p", where "-p" stands for prioritization.

**Skill Discovery with MUSIC and DIAYN:** One of the relevant works on unsupervised RL, DI-AYN (Eysenbach et al., 2019), introduces an information-theoretical objective $\mathcal{F}_{\text{DIAYN}}$, which learns diverse discriminable skills indexed by the latent variable $Z$, mathematically, $\mathcal{F}_{\text{DIAYN}} = I(S; Z) + \mathcal{H}(A \mid S, Z)$. The first term, $I(S; Z)$, in the objective, $\mathcal{F}_{\text{DIAYN}}$, is implemented via a skill discriminator, which serves as a variational lower bound of the original objective (Barber & Agakov, 2003; Eysenbach et al., 2019). The skill discriminator assigns high rewards to the agent, if it can predict the skill-options, $Z$, given the states, $S$. Here, we substitute the full state $S$ with the surrounding state $S^s$ to encourage the agent to learn control skills. DIAYN and MUSIC can be combined as follows: $\mathcal{F}_{\text{MUSIC+DIAYN}} = I(S^a; S^s) + I(S^s; Z) + \mathcal{H}(A \mid S, Z)$. The combined version enables the agent to learn diverse control primitives via skill-conditioned policy (Eysenbach et al., 2019) in an unsupervised fashion.

**Comparison and Combination with DISCERN:** Another relevant work is Discriminative Embedding Reward Networks (DISCERN) (Warde-Farley et al., 2019), whose objective is to maximize the MI between the state $S$ and the goal $G$, namely $I(S; G)$. While MUSIC's objective is to maximize the MI between the agent state $S^a$ and the surrounding state $S^s$, namely $I(S^a; S^s)$. Intuitively, DISCERN attempts to reach a particular goal in each episode, while our method tries to manipulate the surrounding state to *any* different value. MUSIC and DISCERN can be combined as $\mathcal{F}_{\text{MUSIC+DISCERN}} = I(S^a; S^s) + I(S; G)$. Optionally, we can replace the full states $S$ with $S^s$, since it performs better than with $S$ empirically. Through this combination, MUSIC helps DISCERN to learn its discriminative objective.

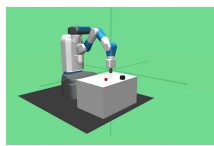 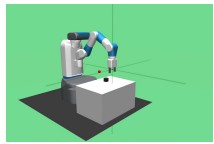 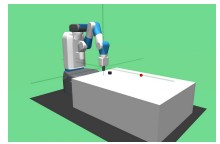 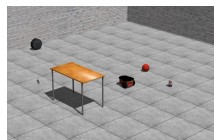

Figure 2: Fetch robot arm manipulation tasks in OpenAI Gym and a navigation task based on the Gazebo simulator: `FetchPush`, `FetchPickAndPlace`, `FetchSlide`, `SocialBot-PlayGround`.

## 4 EXPERIMENTS

**Environments:** To evaluate the proposed methods, we used the robotic manipulation tasks and a navigation task, see Figure 2 (Brockman et al., 2016; Plappert et al., 2018). The navigation task is based on the Gazebo simulator. In the navigation task, the task reward is 1 if the agent reaches the ball, otherwise, the task reward is 0. Here, the agent state is the robot car position and the surrounding state is the red ball. The manipulation environments, including push, pick-and-place, and slide, have a set of predefined goals, which are represented as the red dots. The task for the RL agent is to manipulate the object to the goal positions. In the manipulation task, the agent state is the gripper position and the surrounding state is the object position.

**Experiments:** First, we analyze the control behaviors learned purely with the intrinsic reward, i.e., MUSIC-u. Secondly, we show that the pretrained models can be used for improving performance in conjunction with the task rewards. Interestingly, we show that the pretrained MI estimator can be transferred among different tasks and still improve performance. We compared MUSIC with other methods, including DDPG (Lillicrap et al., 2016), SAC (Haarnoja et al., 2018), DIAYN (Eysenbach et al., 2019), DISCERN (Warde-Farley et al., 2019), PER (Schaul et al., 2016), VIME (Houthooft et al., 2016), ICM (Pathak et al., 2017), and Empowerment (Mohamed & Rezende, 2015). Thirdly, we show some insights about how the MUSIC rewards are distributed across a trajectory. The experimental details are shown in Appendix G. Our code is available at https://github.com/ruizhaogit/music and https://github.com/ruizhaogit/alf.

**Question 1.** *What behavior does MUSIC-u learn?*

We tested MUSIC-u in the robotic manipulation tasks. During training, the agent only receives the intrinsic MUSIC reward. In all three environments, the behavior of reaching objects emerges. In the push environments, the agent learns to push the object around on the table. In the slide environment, the agent learns to slide the object to different directions. Perhaps surprisingly, in the pick-and-place environment, the agent learns to pick up the object from the table without any task reward. All the observations are shown in the supplementary video.

**Question 2.** *How does MUSIC-u compare to Empowerment or ICM?*

We tested our method in the navigation task. We combined our method with PPO (Schulman et al., 2017) and compared the performance with ICM (Pathak et al., 2017) and Empowerment (Mohamed & Rezende, 2015). During training, we only used one of the intrinsic rewards such as MUSIC, ICM, or Empowerment to train the agent. Then, we used the averaged task reward as the evaluation metric. The experimental results are shown in Figure 3 (left). The y-axis represents the mean task reward and the x-axis denotes the training epochs. Figure 3 (right) shows that the MUSIC reward signal $I(S^a, S^s)$ is relatively strong compared to the Empowerment reward signal $I(A, S^s)$. Subsequently, high MI reward encourages the agent to explore more states with higher MI. A theoretical connection between Empowerment and MUSIC is shown in Appendix B. The video starting from 1:28 shows the learned navigation behaviors.

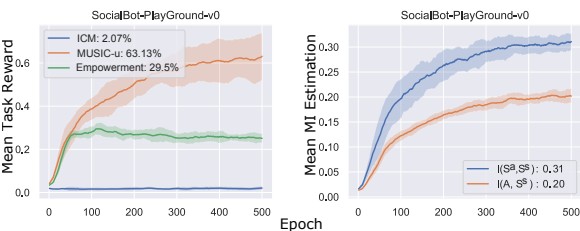

Figure 3: **Experimental results**

**Question 3.** *How does MUSIC compare to DIAYN?*

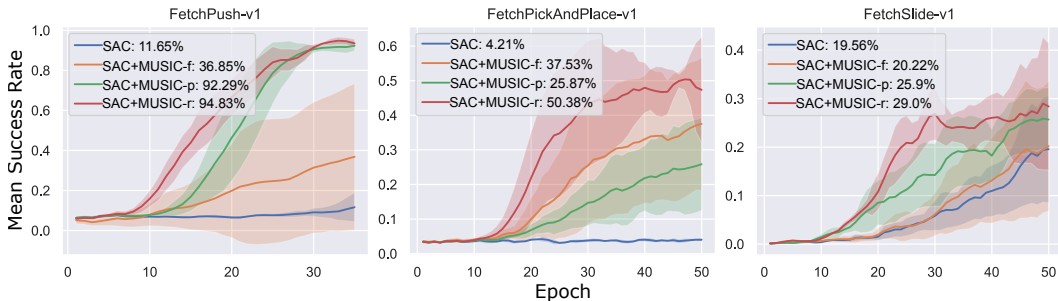

Figure 4: **Mean success rate with standard deviation:** The percentage values after colon (:) represent the best mean success rate during training. The shaded area describes the standard deviation. A full comparison is shown in Appendix D Figure 9.

We compared MUSIC, DIAYN and MUSIC+DIAYN in the pick-and-place environment. For implementing MUSIC+DIAYN, we first pre-train the agent with only MUSIC, and then fine-tune the policy with DIAYN. After pre-training, the MUSIC-trained agent learns manipulation behaviors such as, reaching, pushing, sliding, and picking up an object. Compared to MUSIC, the DIAYN-trained agent rarely learns to pick up the object. It mostly pushes or flicks the object with the gripper. However, the combined model, MUSIC+DIAYN, learns to pick up the object and moves it to different locations, depending on the skill-option. These observations are shown in the video starting from 0:46. From this experiment, we can see that MUSIC helps the agent to learn the DIAYN objective. DIAYN alone doesn't succeed because DIAYN doesn't start to learn any skills until it touches the object, which is rare in the first place. This happens because the skill discriminator only encourages the skills to be different.

**Question 4.** *How does MUSIC+DISCERN compare to DISCERN?*

The combination of MUSIC and DISCERN, encourages the agent to learn to control the object via MUSIC and then move the object to the target position via DISCERN. Table 1 shows that DISCERN+MUSIC significantly outperforms DISCERN.

Table 1: **Comparison of DISCERN with and without MUSIC**

| Method | Push (%) | Pick & Place (%) |
| --- | --- | --- |
| DISCERN | $7.94\% \pm 0.71\%$ | $4.23\% \pm 0.47\%$ |
| R (Task Reward) | $11.65\% \pm 1.36\%$ | $4.21\% \pm 0.46\%$ |
| R+DISCERN | $21.15\% \pm 5.49\%$ | $4.28\% \pm 0.52\%$ |
| R+DISCERN+MUSIC | $95.15\% \pm 8.13\%$ | $48.91\% \pm 12.67\%$ |

This is because that MUSIC emphases more on state-control and teaches the agent to interact with an object. Afterwards, DISCERN teaches the agent to move the object to the goal position in each episode.

**Question 5.** *How can we use MUSIC to accelerate learning?*

We investigated three ways, including MUSIC-f, MUSIC-p, and MUSIC-r, of using MUSIC to accelerate learning in addition to the task reward. We combined these three variants with DDPG and SAC and tested them in the multi-goal robotic tasks. From Figure 4, we can see that all these three methods, including MUSIC-f, MUSIC-p, and MUSIC-r, accelerate learning in the presence of task rewards. Among these variants, the MUSIC-r has the best overall improvements. In the push and pick-and-place tasks, MUSIC enables the agent to learn in a short period of time. In the slide tasks, MUSIC-r also improves the performances by a decent margin.

We also compare our methods with their closest related methods. To be more specific, we compare MUSIC-f against the parameter initialization using DIAYN (Eysenbach et al., 2019); MUSIC-p against Prioritized Experience Replay (PER), which uses TD-errors for prioritization (Schaul et al., 2016); and MUSIC-r versus Variational Information Maximizing Exploration (VIME) (Houthooft et al., 2016). The experimental results are shown in Figure 5. From Figure 5 (1$^{st}$ column), we can see that MUSIC-f enables the agent to learn, while DIAYN does not. In the 2$^{nd}$ column of Figure 5, MUSIC-r performs better than VIME. This result indicates that the MI between states is a crucial quantity for accelerating learning. The MI intrinsic rewards boost performance significantly compared to VIME. This observation is consistent with the experimental results of MUSIC-p and

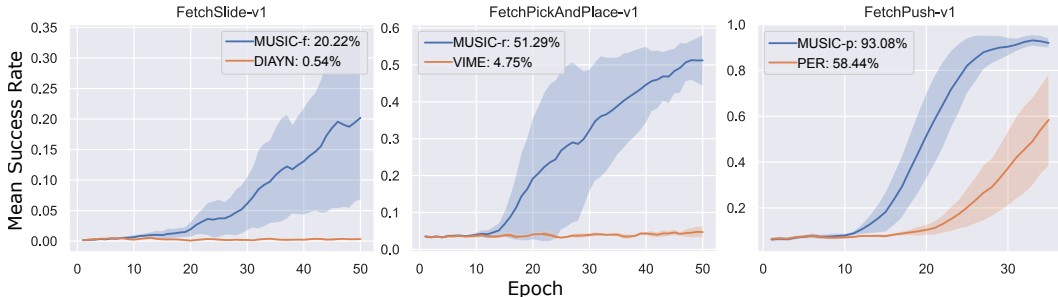

Figure 5: **Performance comparison:** We compare the MUSIC variants, including MUSIC-f, MUSIC-r, and MUSIC-p, with DIAYN, VIME, and PER, respectively. A full comparison is shown in Appendix D Figure 10.

PER, as shown in Figure 5 (3rd column), where the MI-based prioritization framework performs better than the TD-error-based approach, PER. On all tasks, MUSIC enables the agent to learn the benchmark task more quickly.

**Question 6.** *Can the learned MI estimator be transferred to new tasks?*

It would be beneficial if the pretrained MI estimator could be transferred to a new task and still improve the performance (Pan et al., 2010; Bengio, 2012). To verify this idea, we directly applied the pretrained MI estimator from the pick-and-place environment to the push and slide environments, respectively, and train the agent from scratch.

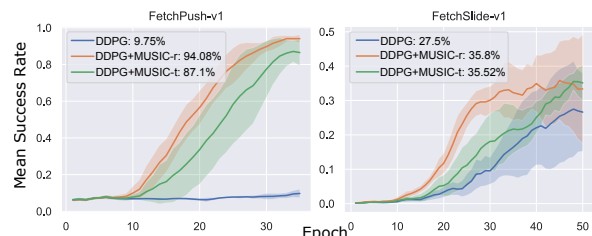

Figure 6: **Transferred MUSIC**

We denote this transferred method as "MUSIC-t", where "-t" stands for transfer. The MUSIC reward function trained in its corresponding environments is denoted as "MUSIC-r". We compared the performances of DDPG, MUSIC-r, and MUSIC-t. The results are in Figure 6, which shows that the transferred MUSIC still improved the performance significantly. Furthermore, as expected, MUSIC-r performed better than MUSIC-t. We can see that the MI estimator can be trained in a task-agnostic (Finn et al., 2017) fashion and later utilized in unseen tasks.

**Question 7.** *How does MUSIC distribute rewards over a trajectory?*

To understand why MUSIC works, we visualize the learned MUSIC-u reward in Figure 7. We can observe that the MI reward peaks between the 4th and 5th frame, where the robot quickly picks up the cube from the table. Around the peak reward value, the middle range reward values are corresponding to the relatively slow move-

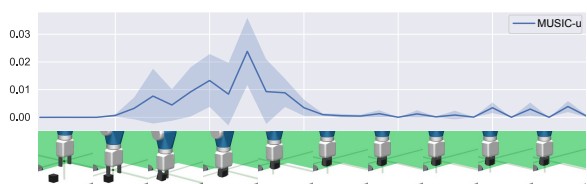

Figure 7: **MUSIC rewards over a trajectory**

ment of the object and the gripper (see the 3rd, 9th, and 10th frame). When there is no contact between the gripper and the cube (see the 1st & 2nd frames), or the gripper holds the object still (see the 6th to 8th frames) the intrinsic reward remains nearly zero. From this example, we see that MUSIC distributes positive intrinsic rewards when the surrounding state has correlated changes with the agent state.

**Question 8.** *How does MUSIC reward compare to reward shaping?*

Here, we want to compare MUSIC and reward shaping and show that MUSIC cannot be easily replaced by reward shaping. We consider a simple L2-norm reward shaping, which is the distance between the robot's gripper and the object. With this hand-engineered reward, the agent learns to move its gripper close to the object but barely touch the object. However, with MUSIC reward, the

agent reaches the object and moves it into different locations. MUSIC automatically induces a lot of hand-engineered rewards, including the L2-norm distance reward between the gripper and the object, the contact reward between the agent and the object, the L2-norm distance reward between the object and the goal position and any other rewards that maximize the mutual information between the agent and the surrounding state. From this perspective, MUSIC can be considered as a meta-reward for the state-control tasks, which helps the agent to learn any specific downstream tasks that falls into this category.

**Question 9.** *Can MUSIC help the agent to learn when there are multiple surrounding objects?*

When there are multiple objects, the agent is trained to maximize the MI between the surrounding objects and the agent via MUSIC. In the case that there is a red and a blue ball on the ground, with MUSIC, the agent learns to reach both balls and sometimes also learns to use one ball to hit the other ball. The results are shown in the supplementary video starting from 1:56.

**Summary and Future Work:** We can see that, with different combinations of the surrounding state and the agent state, the agent is able to learn different control behaviors. We can train a skill-conditioned policy corresponding to different combinations of the agent state and the surrounding state and later use the pretrained policy for the tasks at hand, see Appendix F "Skill Discovery for Hierarchical Reinforcement Learning". In some cases, when there is no clear agent-surrounding separation or the existing separation is suboptimal, new methods are needed to divide and select the states automatically. Another future work direction is to extend the current method to the partially observed cases. For example, we can combine MUSIC with state estimation methods and extend MUSIC to the partially observed settings.

## 5 RELATED WORK

Intrinsically motivated RL is a challenging topic. We divide the previous works in three categories. In the first category, intrinsic rewards are often used to help the agent learn more efficiently to solve tasks. For example, Jung et al. (2011) and Mohamed & Rezende (2015) use empowerment, which is the channel capacity between states and actions. A theoretical connection between MU-SIC and empowerment is shown in Appendix B. VIME (Houthooft et al., 2016) and ICM (Pathak et al., 2017) use curiosity as intrinsic rewards to encourage the agents to explore the environment more thoroughly. Another category of work on intrinsic motivation for RL is to discover meaningful skills, such as Variational Intrinsic Control (VIC) (Gregor et al., 2016), DIAYN (Eysenbach et al., 2019), and Explore Discover Learn (EDL) (Campos et al., 2020). In the third category, intrinsic motivation also helps the agent to learn goal-conditioned policies. Warde-Farley et al. (2019) proposed DISCERN, a method to learn a MI objective between the states and goals. Based on DIS-CERN, Pong et al. (2019) introduced Skew-fit, which adapts a maximum entropy strategy to sample goals from the replay buffer (Zhao et al., 2019) in order to make the agent learn more efficiently in the absence of rewards. However, these methods fail to enable the agent to learn meaningful interaction skills in the environment, such as in the robot manipulation tasks. Our work is based on the agent-surrounding separation concept and drives an efficient state intrinsic control objective, which empowers RL agents to learn meaningful interaction and control skills without any task reward. A recent work (Song et al., 2020) with similar motivation, introduces mega-reward, which aims to maximize the control capabilities of agents on given entities in a given environment and show promising results in Atari games. Another related work (Dilokthanakul et al., 2019) proposes feature control as intrinsic motivation and shows state-of-the-art results in Montezuma's revenge.

In this paper, we introduce MUSIC, a method that uses the MI between the surrounding state and the agent state as the intrinsic reward. In contrast to previous works on intrinsic rewards (Mohamed & Rezende, 2015; Houthooft et al., 2016; Pathak et al., 2017; Eysenbach et al., 2019; Warde-Farley et al., 2019), MUSIC encourages the agent to interact with the interested part of the environment, which is represented by the surrounding state, and learn to control it. The MUSIC intrinsic reward is critical when controlling a specific subset of the environmental state is the key to complete the task, such as the case in robotic manipulation tasks. Our method is complementary to these previous works, such as DIAYN and DISCERN, and can be combined with them. Inspired by previous works (Schaul et al., 2016; Houthooft et al., 2016; Eysenbach et al., 2019), we additionally demonstrate three variants, including MUSIC-based fine-tuning, rewarding, and prioritizing mechanisms, to significantly accelerate learning in the downstream tasks.

## 6 CONCLUSION

This paper introduces Mutual Information-based State Intrinsic Control (MUSIC), an unsupervised RL framework for learning useful control behaviors. The derived efficient MI-based theoretical objective encourages the agent to control states without any task reward. MUSIC enables the agent to self-learn different control behaviors, which are non-trivial, intuitively meaningful, and useful for learning and planning. Additionally, the pretrained policy and the MI estimator significantly accelerate learning in the presence of task rewards. We evaluated three MUSIC-based variants in different environments and demonstrate a substantial improvement in learning efficiency compared to state-of-the-art methods.

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

## APPENDIX

## A  MONOTONICALLY INCREASING RELATIONSHIP

**Lemma 2.** *There is a monotonically increasing relationship between $I_\phi(S^s; S^a \mid \mathcal{T})$ and $\mathbb{E}_{\mathbb{P}_{\mathcal{T}'}}[I_\phi(S^s; S^a \mid \mathcal{T}')]$, mathematically,*

$$I_\phi(S^s; S^a \mid \mathcal{T}) \Join \mathbb{E}_{\mathbb{P}_{\mathcal{T}'}}[I_\phi(S^s; S^a \mid \mathcal{T}')], \tag{6}$$

*where $S^s$, $S^a$, and $\mathcal{T}$ denote the surrounding state, the agent state, and the trajectory, respectively. The trajectory fractions are defined as the adjacent state pairs, namely $\mathcal{T}' = \{S_t, S_{t+1}\}$. The symbol $\Join$ denotes a monotonically increasing relationship between two variables and $\phi$ represents the parameter of the statistics model in MINE.*

*Proof.* The derivation of the monotonically increasing relationship is shown as follows:

$$I_\phi(S^s; S^a \mid \mathcal{T}) = \mathbb{E}_{\mathbb{P}_{S^s S^a \mid \mathcal{T}}}[T_\phi] - \log(\mathbb{E}_{\mathbb{P}_{S^s \mid \mathcal{T}} \otimes \mathbb{P}_{S^a \mid \mathcal{T}}}[e^{T_\phi}]) \tag{7}$$

$$\Join \mathbb{E}_{\mathbb{P}_{S^s S^a \mid \mathcal{T}}}[T_\phi] - \mathbb{E}_{\mathbb{P}_{S^s \mid \mathcal{T}} \otimes \mathbb{P}_{S^a \mid \mathcal{T}}}[e^{T_\phi}] \tag{8}$$

$$= \mathbb{E}_{\mathbb{P}_{\mathcal{T}'}}[\mathbb{E}_{\mathbb{P}_{S^s S^a \mid \mathcal{T}'}}[T_\phi] - \mathbb{E}_{\mathbb{P}_{S^s \mid \mathcal{T}'} \otimes \mathbb{P}_{S^a \mid \mathcal{T}'}}[e^{T_\phi}]] \tag{9}$$

$$\Join \mathbb{E}_{\mathbb{P}_{\mathcal{T}'}}[\mathbb{E}_{\mathbb{P}_{S^s S^a \mid \mathcal{T}'}}[T_\phi] - \log(\mathbb{E}_{\mathbb{P}_{S^s \mid \mathcal{T}'} \otimes \mathbb{P}_{S^a \mid \mathcal{T}'}}[e^{T_\phi}])] = \mathbb{E}_{\mathbb{P}_{\mathcal{T}'}}[I_\phi(S^s; S^a \mid \mathcal{T}')], \tag{10}$$

where $T_\phi$ represents a neural network, whose inputs are state samples and the output is a scalar. For simplicity, we use the symbol $\Join$ to denote a monotonically increasing relationship between two

variables, for example, $\log(x) \ltimes x$ means that as the value of $x$ increases, the value of $\log(x)$ also increases and vice versa. To decompose the lower bound Equation (7) into small parts, we make the following derivations, see Equation (8,9,10). Deriving from Equation (7) to Equation (8), we use the property that $\log(x) \ltimes x$. Here, the new form, Equation (8), allows us to decompose the MI estimation into the expectation over MI estimations of each trajectory fractions, Equation (9). To be more specific, we move the implicit expectation over trajectory fractions in Equation (8) to the front, and then have Equation (9). The quantity inside the expectation over trajectory fractions is the MI estimation using only each trajectory fraction, see Equation (9). We use the property, $\log(x) \ltimes x$, again to derive from Equation (9) to Equation (10). □

## B    CONNECTION TO EMPOWERMENT

The state $S$ contains the surrounding state $S^s$ and the agent state $S^a$. For example, in robotic tasks, the surrounding state and the agent state represents the object state and the end-effector state, respectively. The action space is the change of the gripper position and the status of the gripper, such as open or closed. Note that, the agent's action directly alters the agent state.

Here, given the assumption that the transform, $S^a = F(A)$, from the action, $A$, to the agent state, $S^a$, is a smooth and uniquely invertible mapping (Kraskov et al., 2004), then we can prove that the MUSIC objective, $I(S^a, S^s)$, is equivalent to the empowerment objective, $I(A, S^s)$.

The empowerment objective (Klyubin et al., 2005; Salge et al., 2014; Mohamed & Rezende, 2015) is defined as the channel capacity in information theory, which means the amount of information contained in the action $A$ about the state $S$, mathematically:

$$\mathcal{E} = I(S, A). \tag{11}$$

Here, we replace the state variable $S$ with the surrounding state $S^s$, we have the empowerment objective as follows,

$$\mathcal{E} = I(S^s, A). \tag{12}$$

**Theorem 3.** *The MUSIC objective, $I(S^a, S^s)$, is equivalent to the empowerment objective, $I(A, S^s)$, given the assumption that the transform, $S^a = F(A)$, is a smooth and uniquely invertible mapping:*

$$I(S^a, S^s) = I(A, S^s) \tag{13}$$

where $S^s$, $S^a$, and $A$ denote the surrounding state, the agent state, and the action, respectively.

*Proof.*

$$I(S^a, S^s) = \int \int ds^a ds^s p(s^a, s^s) \log \frac{p(s^a, s^s)}{p(s^a)p(s^s)} \tag{14}$$

$$= \int \int ds^a ds^s \left\| \frac{\partial A}{\partial S^a} \right\| p(a, s^s) \log \frac{\left\| \frac{\partial A}{\partial S^a} \right\| p(a, s^s)}{\left\| \frac{\partial A}{\partial S^a} \right\| p(a)p(s^s)} \tag{15}$$

$$= \int \int ds^a ds^s J_A(s^a) p(a, s^s) \log \frac{J_A(s^a)p(a, s^s)}{J_A(s^a)p(a)p(s^s)} \tag{16}$$

$$= \int \int da ds^s p(a, s^s) \log \frac{p(a, s^s)}{p(a)p(s^s)} \tag{17}$$

$$= I(A, S^s) \tag{18}$$

□

## C    MUTUAL INFORMATION NEURAL ESTIMATOR TRAINING

---

**Algorithm 2:** MINE (Belghazi et al., 2018)

---

$\theta \leftarrow$ initialize network parameters

**repeat**

    Draw $b$ minibatch samples from the joint distribution:

    $(\boldsymbol{x}^{(1)}, \boldsymbol{z}^{(1)}), \ldots, (\boldsymbol{x}^{(b)}, \boldsymbol{z}^{(b)}) \sim \mathbb{P}_{XZ}$

    Draw $n$ samples from the $Z$ marginal distribution:

    $\bar{\boldsymbol{z}}^{(1)}, \ldots, \bar{\boldsymbol{z}}^{(b)} \sim \mathbb{P}_Z$

    Evaluate the lower-bound:

    $\mathcal{V}(\theta) \leftarrow \frac{1}{b}\sum_{i=1}^{b} T_\phi(\boldsymbol{x}^{(i)}, \boldsymbol{z}^{(i)}) - \log(\frac{1}{b}\sum_{i=1}^{b} e^{T_\phi(\boldsymbol{x}^{(i)}, \bar{\boldsymbol{z}}^{(i)})})$

    Evaluate bias corrected gradients (e.g., moving average):

    $\widehat{G}(\theta) \leftarrow \widetilde{\nabla}_\theta \mathcal{V}(\theta)$

    Update the statistics network parameters:

    $\theta \leftarrow \theta + \widehat{G}(\theta)$

**until** convergence

---

One potential pitfall of training the RL agent using the MINE reward is that the MINE reward signal can be relatively small compared to the task reward signal. The practical guidance to solve this problem is to tune the scale of the MINE reward to be similar to the scale of the task reward.

## D    EXPERIMENTAL RESULTS

The learned control behaviors without supervision are shown in Figure 8 as well as in the supplementary video. The detailed experimental results are shown in Figure 9 and Figure 10.

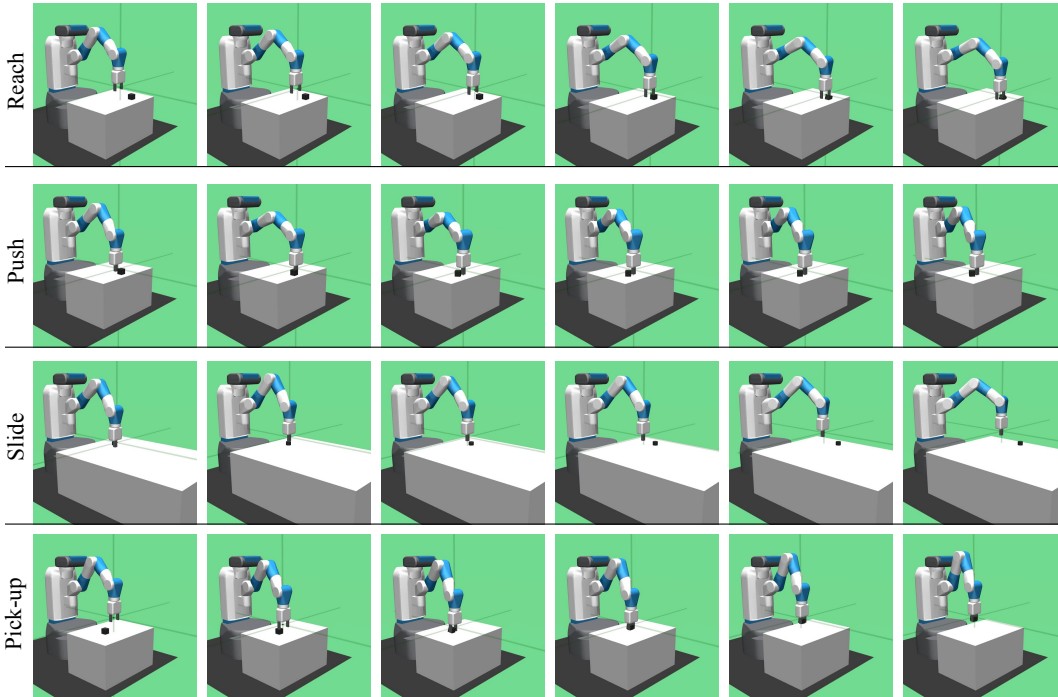

Figure 8: **Learned Control behaviors with MUSIC**: Without any reward, MUSIC enables the agent to learn control behaviors, such as reaching, pushing, sliding, and picking up an object. The learned behaviors are shown in the supplementary video.

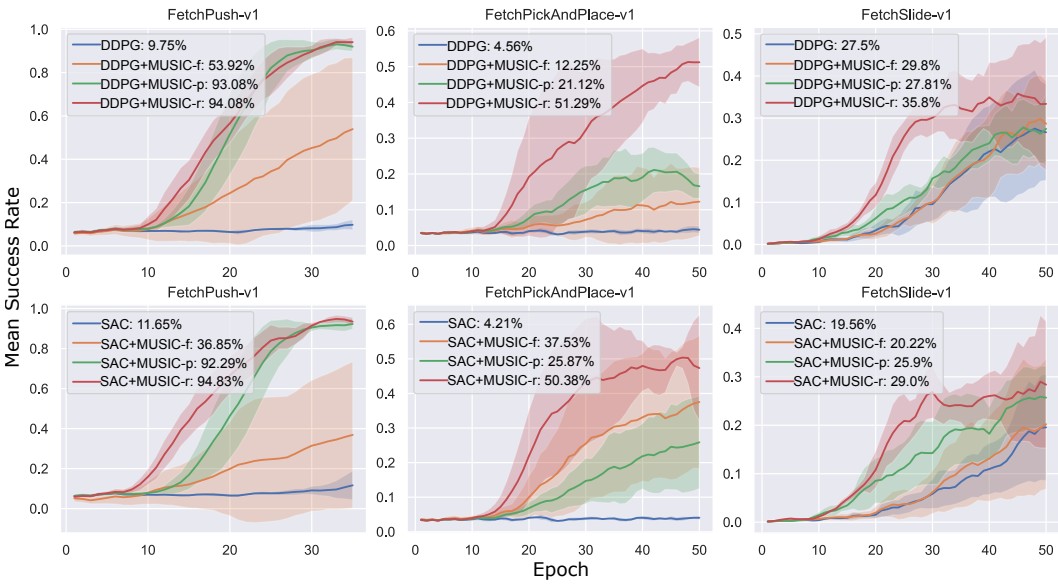

Figure 9: **Mean success rate with standard deviation:** The percentage values after colon (:) represent the best mean success rate during training. The shaded area describes the standard deviation.

## E  COMPARISON OF VARIATIONAL MI-BASED AND MINE-BASED IMPLEMENTATIONS

Here, we compare the variational approach-based (Barber & Agakov, 2003) implementation of MUSIC and MINE-based implementation (Belghazi et al., 2018) of MUSIC in Table 2. All the experiments are conducted with 5 different random seeds. The performance metric is mean success rate (%) ± standard deviation. The "Task-r" stands for the task reward. From Table 2, we can see that

Table 2: **Comparison of variational MI (v)-based and MINE (m)-based MUSIC**

| Method | Push (%) | Pick & Place (%) |
|---|---|---|
| Task-r+MUSIC(v) | 94.9% ± 5.83% | 49.17% ± 4.9% |
| Task-r+MUSIC(m) | 94.83% ± 4.95% | 50.38% ± 8.8% |

the performance of these two MI estimation methods are similar. However, the variational method introduces additional complicated sampling mechanisms, and two additional hyper-parameters, i.e., the number of the candidates and the type of the similarity measurement (Barber & Agakov, 2003; Eysenbach et al., 2019; Warde-Farley et al., 2019). In contrast, MINE-style MUSIC is easier to implement and has less hyper-parameters to tune. Furthermore, the derived objective improves the scalability of the MINE-style MUSIC.

## F  SKILL DISCOVERY FOR HIERARCHICAL REINFORCEMENT LEARNING

In this section, we explore the direction of Hierarchical Reinforcement Learning based on MUSIC.

For example, in the Fetch robot arm pick-and-place environment, we have the follow states as the observation: `grip_pos`, `object_pos`, `object_velp`, `object_rot`, `object_velr`, where the abbreviation "`pos`" stands for position; "`rot`" stands for rotation; "`velp`" stands for linear velocity, and "`velr`" stands for rotational velocity.

The `grip_pos` is the agent state. The surrounding states are `object_pos`, `object_velp`, `object_rot`, `object_velr`. In Table 3, we show the MI value with different state-pair combinations prior to training and post to training. When the MI value difference is high, it means that

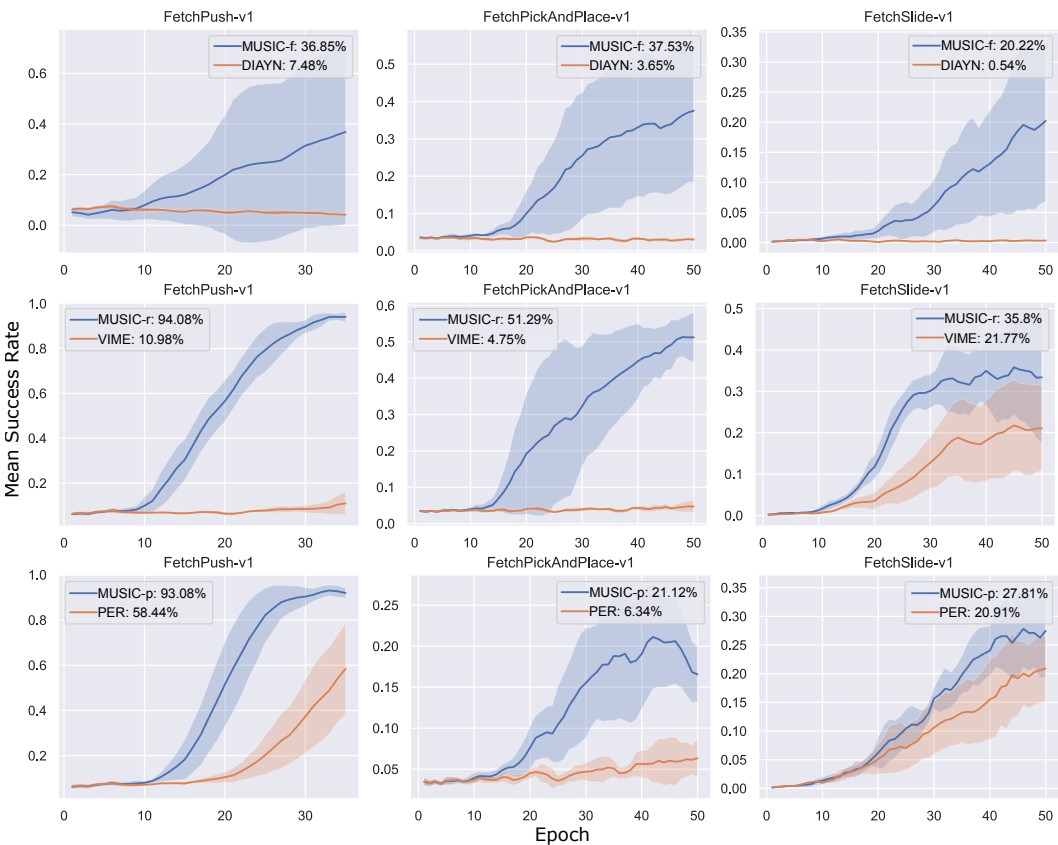

Figure 10: **Performance comparison:** We compare the MUSIC variants, including MUSIC-f, MUSIC-r, and MUSIC-p, with DIAYN, VIME, and PER, respectively.

the agent has a good learning progress with the corresponding MI objective. From Table 3 first row,

Table 3: **Mutual Information estimation prior and post to the training**

| Mutual Information Objective | Prior-train Value | Post-train Value |
|---|---|---|
| MI(grip_pos; object_pos) | $0.003 \pm 0.017$ | $0.164 \pm 0.055$ |
| MI(grip_pos; object_rot) | $0.017 \pm 0.084$ | $0.461 \pm 0.088$ |
| MI(grip_pos; object_velp) | $0.005 \pm 0.010$ | $0.157 \pm 0.050$ |
| MI(grip_pos; object_velr) | $0.016 \pm 0.083$ | $0.438 \pm 0.084$ |

we can see that with the intrinsic reward MI(grip_pos; object_pos), the agent achieves a high MI after training, which means that the agent learns to better control the object positions using its gripper. Similarly, in the second row of the table, with MI(grip_pos; object_rot), the agent learns to control object rotation with its gripper.

From the experimental results, we can see that with different combination of state-pairs of the agent and surrounding state, the agent can learn different skills, such as manipulate object positions or rotations. We can connect these learned skills with different skill-options (Eysenbach et al., 2019) and train a meta-controller to control these motion primitives to complete long-horizon tasks in a hierarchical reinforcement learning framework (Eysenbach et al., 2019). We consider this as a future research direction, which could be a solution in solving more challenging and complex long-horizon tasks.

## G EXPERIMENTAL DETAILS

We ran all the methods in each environment with 5 different random seeds and report the mean success rate and the standard deviation. The experiments of the robotic manipulation tasks in this paper use the following hyper-parameters:

- Actor and critic networks: 3 layers with 256 units each and ReLU non-linearities
- Adam optimizer (Kingma & Ba, 2014) with $1 \cdot 10^{-3}$ for training both actor and critic
- Buffer size: $10^6$ transitions
- Polyak-averaging coefficient: 0.95
- Action L2 norm coefficient: 1.0
- Observation clipping: $[-200, 200]$
- Batch size: 256
- Rollouts per MPI worker: 2
- Number of MPI workers: 16
- Cycles per epoch: 50
- Batches per cycle: 40
- Test rollouts per epoch: 10
- Probability of random actions: 0.3
- Scale of additive Gaussian noise: 0.2
- Scale of the mutual information reward: 5000

The specific hyper-parameters for DIAYN are follows:

- Number of skill options: 5
- Discriminate skills based on the surrounding state

The specific hyper-parameters for VIME are follows:

- Weight for intrinsic reward $\eta$: 0.2
- Bayesian Neural Network (BNN) learning rate: 0.0001
- BNN number of hidden units: 32
- BNN number of layers: 2
- Prior standard deviation: 0.5
- Use second order update: `True`
- Use information gain: `True`
- Use KL ratio: `True`
- Number updates per sample: 1

The specific hyper-parameters for DISCERN are follows:

- Number of candidates to calculate the contrastive loss: 10
- Calculate the MI using the surrounding state

The specific hyper-parameters for PER are follows:

- Prioritization strength $\alpha$: 0.6
- Importance sampling factor $\beta$: 0.4

The specific hyper-parameter for SAC is following:

- Weight of the entropy reward: 0.02

