# OpenReview forum: "Mutual Information State Intrinsic Control"
_ICLR.cc/2021/Conference — ICLR 2021 Spotlight_

### Official Review · AnonReviewer4 · 2020-10-22
**Simple intuition, impressive results, great paper**

**Rating:** 8
**Confidence:** 5

**Review:**

This work introduces MUSIC, a framework for intrinsically motivated RL, where the intrinsic reward comes from maximizing the mutual information between the agent's state and the surrounding environment's state. The authors motivate and describe this approach, explain its incorporation into various training modes, exhaustively characterize its properties, and compare to numerous related past approaches. Although it is somewhat specific to particular environment domains, MUSIC offers a compelling addition to the family of intrinsically motivated RL algorithms based on concepts of mutual information.

### Clarity
This paper is, for the most part, a model of clarity. The proposed algorithm follows from a clear intuition and the results present a detailed and organized characterization/validation. The authors could further improve clarity by adding a bit more exposition around the training of the mutual information estimator. This is only a minor issue, since the mutual information framework itself is well established by now. Still, it would help to more concretely describe the implementation details covered in Section 3.2, perhaps in another small section of the Appendix.

### Quality
This paper is very high quality. The experiments are thorough and well organized, addressing an impressive number of (literally enumerated) questions. This also serves to demonstrates an impressive versatility of MUSIC, the proposed technique, while simultaneously enabling comparison to a **wide** range of past methods and showcasing a variety of potential uses. However, there is a lack of guidance around practical challenges facing this technique and its potential pitfalls.  For example, Campos et al. (_ICML_ 2020) describe some failure modes associated with simultaneously learning networks for estimating mutual information and using them to train policies. It would be valuable to know if MUSIC has similar (or otherwise noteworthy) failure modes.

### Originality and Significance
As the paper describes, there is a large body of work on intrinsically motivated RL as well as intrinsic rewards derived from mutual information-based objectives. In many cases, the types of behaviors those alternative approaches would hope to encourage are the same as those learned by MUSIC. From what the paper demonstrates, it seems like MUSIC is a more successful iteration of these attempts. That is not meant to diminish its significance -- it's a very hard problem! Ostensibly, MUSIC incorporates the right inductive bias through its decomposition of the state. I expect that the field will find both the technique to be useful as well as the general insights brought about through this work. My only concern regarding the significance is whether MUSIC is only applicable to a relatively narrow set of domains. Even if we assume that the state can be cleanly decomposed into the agent/surrounding constituents, how might MUSIC handle things like partial observability or other types of uncertainty that may affect estimations of mutual information? The paper may benefit from a brief discussion around generality and, if appropriate, how future work may address any issues therein.

**Pros**
- Simple extension of an existing framework leading to clear and versatile improvements within a challenging problem
- Paper is exceptionally clear and well-organized
- Impressively thorough experimental characterization/validation
- Code is provided to help improve reproducibility and external adoption

**Cons**
- Some gaps in practical guidance and discussion of potential pitfalls
- Generality of the approach (with respect to environment/task setting) is a bit unclear

---

> ### Author Response · Authors · 2020-11-16
> **To AnonReviewer4**
>
> Thank you for your valuable feedback!
>
> - One potential pitfall is that the MI reward signal might be relatively small compared to the task reward. The practical guidance to solve this problem would be to tune the scale of the MI reward to be similar to the scale of the task reward. If the scale of the rewards does not match, for example, the MI reward is always around 0.0001 and the task reward is mostly around 1, then the performance of the experiments would be affected. We added this in Appendix C of the updated paper. We also added more exposition about training the mutual information estimator in Appendix C.
>
> - About generality of the approach:
>
> 1. Our method is based on the agent-surrounding separation. In most RL setting, there is a natural separation between the agent and the environment. In the case there is no clear separation, or, the existing separation is suboptimal, then the automatic methods are needed to divide and select the states. We think that future works in this direction would be interesting. We added this discussion in the future work section of the updated paper.
>
> 2. In the partially observed settings, we can combine our method with state estimation methods, which estimate the state from multiple historical observations. For example, imagine a robot is equipped with a camera, our method potentially can be applied to train policies via maximizing the MI between the agent state and the surrounding state inferred using state estimation methods. We think this is a very interesting direction for future work.

---

### Official Review · AnonReviewer1 · 2020-10-26
**State Control as Intrinsic Motivation**

**Rating:** 7
**Confidence:** 5

**Review:**

## Summary
This paper proposes the use of state control as intrinsic motivation. It does so by separating the overall state into an agent state and a surrounding state. The idea then is to maximize the mutual information between the agent's internal state and the environment state. This mutual information is given as a reward to the agent which it attempts to maximize using policy gradient algorithms. Experiments attempt to validate the usefulness of such an intrinsic reward signal for pre-training an agent, or for augmenting a given task reward.

## Positives
+ The proposed idea of tying an agent's internal state to the surrounding environment is an interesting new metric. A practical use-case for embodiment is extremely interesting and this paper champions such a use-case fairly well.
+ Adapting the use of MINE for computing the MUSIC reward is a useful contribution.
+ Experiments sufficiently showcase the viability of the idea in various manipulation tasks.
+ The video showcasing the various behaviors learned using MUSIC rewards in different environments, as well as the different skill learned when combined with DIAYN give a clear understanding of the effect of this intrinsic reward.
+ Comparison with various exploration bonuses, skill learning priors, and prioritization schemes gives a clear idea of how MUSIC compares with other similar techniques.

## Drawbacks:
- While I find the idea and the experiments quite interesting, I do feel there is some improvement necessary in presenting the idea and comparing it with the idea that is most related to MUSIC, which is empowerment.
- My main gripe with the paper is that the background of mutual information, estimating mutual information using MINE, and the adaptation in this paper in order to compute mutual information for agent trajectory is not sufficient or clear. While an interested reader can go to the referenced work and glean more information, the idea itself should be sufficiently understood by reading this paper. I would argue this is not true with the current state of the draft.
- While there is an experiment for comparison with empowerment, more explanation in the related work is necessary for a reader unfamiliar with that body of work.
- While the paper deals with the idea that an agent uses its knowledge of itself and the environment to affect the environment as much as possible, it does not discuss sufficiently how such an agent-environment divide might be possible or not possible in various scenarios where RL would be a useful solution mechanism. For example, would it be possible to use MUSIC in Atari, or for optimizing a recommendation system? From my understanding, it would not be straightforward. But this is a question that seems like it deserves more discussion in the paper.

## Questions:
- It is unclear why computing the mutual information between the agent and surrounding state needs information of the trajectory. Is it just a practical issue, as referenced in Section 3.2?
- Related to the above (and quite possibly answered by it), why are all the states in a trajectory used for computing the MI? Is correlation in the samples not an issue? Why not?
- In the experiments with the Fetch robot, the off-policy algorithm used by HER is labeled as SAC (Figure 4). However, the experimental details in the appendix as well as the code provided with the supplementary material seems to be specific to DDPG. Is there a reason for this discrepancy?
- Where can we expect MUSIC to not provide informative rewards? Will it work in scenarios where agent actions will change the surrounding state in a temporally offset manner? What happens when the surrounding state changes due to some change in agent state, but delayed. For example, in the Fetch Slide task, the agent must slide the block across the table, but after the initial push any movement the robot makes does not affect the block's trajectory. How does this dynamic affect MUSIC rewards, and subsequent performance on the task? I understand if this particular question is out of the scope of the paper. But curious as to the authors' views.

## Conclusion
MUSIC rewards are an interesting idea and would be a good addition to intrinsic motivation literature. A more well-rounded explanation of the techniques used in this paper and the capabilities and use-case for MUSIC would elevate the paper.

---

> ### Author Response · Authors · 2020-11-16
> **To AnonReviewer1**
>
> Thank you for your valuable feedback!
>
> - Thanks for this suggestion! We compared our method with empowerment in Question 2 empirically and in Appendix B theoretically. We will add more discussion and comparison with empowerment in the final version of the paper, to allow people who are unfamiliar with empowerment to understand the differences.
>
> - Thanks for pointing out this issue! For now, we added a Section “Mutual Information Neural Estimator Training” in Appendix C of the updated paper. We will elaborate more on the background of estimating mutual information using MINE in the paper.
>
> - This is a really good point. We agree that in some environments the agent-environment division might not be readily defined. It might be possible to acquire such division by using automatic methods, such as RL, which we didn’t explore in this paper. We include a discussion on this as well as point out the direction in the future work section in the updated paper. In the case of Atari, there is recent work [1], whose idea is quite similar to our work. Song et al. [1] introduce mega-reward, which aims to maximize the control capabilities of agents on given entities in a given environment and show promising results in Atari games.
> Reference:
> [1] Mega-Reward: Achieving Human-Level Play without Extrinsic Rewards. Song et al. 2020 AAAI
>
> To your questions:
>
> - Yes, we consider this as a practical issue of using MINE in RL.
>
> - Sorry for the confusion. We use all the states in a trajectory to train the MI estimator instead of using them to compute the MI reward. We only use the adjacent state pair to calculate the MI reward, see Equation (5) in the paper. In more detail, we define the reward for each transition at a given time-step as the mutual information of the pair of adjacent states at that time-step. However, in practice, we find that it is not very efficient to train the MI estimator using state pairs. To counter this issue, we use all the states in the same trajectory in a batch to train the MI estimator, since more empirical samples help to reduce variance and therefore accelerate learning.
>
> - We implemented SAC on top of DDPG in the codebase. The SAC is activated with the “sac” argument in the scripts, including “config.py”, “actor_critic.py”, and “ddpg.py”. Sorry for the confusion, we will make the codebase clearer.
>
> - We believe that the temporally offset case would not be a big problem for MUSIC because this issue can be naturally mitigated by the Bellman backup. When optimizing for the rewards, the RL algorithm naturally takes the future MUSIC reward into account. We do agree that delayed rewards will make the exploration harder, but that is a general problem with sparse and delayed rewards in RL.

---

> > ### Comment · AnonReviewer1 · 2020-11-18
> > **Response to Authors**
> >
> > - The addition of the MINE algorithm is a good step towards improving presentation. I am looking forward to seeing the discussion on empowerment. In addition to the Song et al. paper, which is a good comparison to add, I would also point the authors to "Explore, Discover and Learn: Unsupervised Discovery of State-Covering Skills", by Campos et al. ICML 2020. Their discussion on the drawbacks of mutual information approaches might be useful in your comparison to mutual information as well.
> > - Apart from amending the code base, I would encourage the authors to also include the SAC hyperparameters used in the experimental details. At the very least, the SAC entropy parameter used should be included.
> > - Thank you for the other responses.

---

> > > ### Author Response · Authors · 2020-11-18
> > > **To AnonReviewer1**
> > >
> > > - Thank you for your response!
> > >
> > > - Thanks for pointing us to the related work "Explore, Discover and Learn: Unsupervised Discovery of State-Covering Skills". This work addresses the limitation that the existing skill-learning algorithms have poor coverage of the state space. Subsequently, the authors propose EDL, a state-covering skill learning algorithm. Our work focuses more on state-control. We believe that EDL is complementary to our work and can be combined with it. We now discuss EDL in the related work section.
> > >
> > > - Thank you for this suggestion! We now add the hyperparameter, the SAC entropy parameter, in Appendix G.

---

### Official Review · AnonReviewer2 · 2020-10-28
**Interesting and well-motivated approach with some slight issues in clarity**

**Rating:** 7
**Confidence:** 3

**Review:**

Summary: This paper introduces MUSIC, a reinforcement learning approach that separates the agent state from its surrounding state and trains the agent to maximize the mutual information between the two. This implies that the agent has control over the surrounding state. The approach is evaluated within four environments and compared to multiple baselines.


This paper is well-motivated and the approach is interesting. The paper is mostly well-written, though I found parts to be somewhat confusing. Code is provided as well as hyperparameters so the approach seems reproducible. The experiments are strong as the approach is evaluated within multiple environments with extensive comparisons to relevant baselines.


MUSIC is shown to achieve very good performance on simulated robotic tasks, and was able to improve performance when combined with other intrinsic reward and RL methods. I think this is an interesting direction and it does make sense to separate out the agent’s state from the environment state. For these reasons I do think the paper should be accepted.


However, I found the description of the methodology in section 3.2 to be very confusing. The equations are referred to before they are introduced which was unexpected. Hence, this section would be greatly improved by some rearranging. I also did not understand what exactly T was. What does this function output and how is it trained?


Comments:


- Some other related works are [1] which uses an intrinsic reward to maximize the controllable entities in the environment and [2] which learns an intrinsic reward that maximizes controllable features.


-  Question 3 in the paper does not refer to any figure (does this correspond to figure 5?). Where are the MUSIC + DIAYN results?


- Is the reward in Question 8 the negative L2 norm?


- How does MUSIC alone perform in Table 1? This should be included here as well.


[1] Mega-Reward: Achieving Human-Level Play without Extrinsic Rewards. Song et al.


[2] Feature Control as Intrinsic Motivation for Hierarchical Reinforcement Learning. Dilokthanakul et al.

---

> ### Author Response · Authors · 2020-11-16
> **To AnonReviewer2**
>
> Thank you for your valuable suggestions!
>
> - To improve clarity, we take your suggestion rearranging the text in section 3.2. Now the equations are referred to after they are introduced.
> T is the statistics model, which is parameterized by a deep neural network, whose inputs are the empirical samples, which are the state samples in our case, and the output is a scalar value, which is later used for calculating the MI estimation in the Donsker-Varadhan representation of the mutual information quantity. The function T is trained to maximize the lower bond of the MI quantity, see Equation (3) in the paper.
>
> - Thank you for pointing us to these two interesting related works [1,2]!
> The recent work [1] introduces mega-reward, which aims to maximize the control capabilities of agents on given entities in a given environment and show promising results in Atari games. Our work bears a similar motivation, which is to encourage the agent to gain control over its surroundings and show promising results in the continuous control domain. Dilokthanakul et al. [2] suggest feature control as intrinsic motivation and show state-of-the-art results in Montezuma’s revenge. We now discussed these two works in the related work section of the updated paper.
> Reference:
> [1] Mega-Reward: Achieving Human-Level Play without Extrinsic Rewards. Song et al. 2020 AAAI
> [2] Feature Control as Intrinsic Motivation for Hierarchical Reinforcement Learning. Dilokthanakul et al. 2017
>
> - Sorry for the confusion. In Question 3, the MUSIC + DIAYN results and observations refer to the supplementary video starting from 0:46, which we plan to include in the paper. Figure 5 corresponds to Question 5.
>
> - Yes, exactly, the reward in Question 8 is the negative L2 norm.
>
> - The performance of MUSIC alone (MUSIC-u, u: unsupervised) is shown in the supplementary video and discussed in Question 1. The reason that we didn’t add the performance of MUSIC-u in Table 1 is that all the other methods in Table are goal-reaching methods. However, MUSIC-u purely explores the world without any goal in mind. Thus, we think it is relatively unfair to compare the performance of these methods with the performance of MUSIC-u in the goal-reaching tasks. Therefore, we reported the performance of the combined version of MUSIC.

---

### Official Review · AnonReviewer3 · 2020-10-29
**Review for "Mutual Information State Intrinsic Control"**

**Rating:** 7
**Confidence:** 3

**Review:**

The paper propose MUSIC, an RL algorithm for learning controllers in a unsupervised way. They key idea of the proposed algorithm is to separate the state of the robot like joint angles from the state of the environments such as location of an external object and optimize the mutual information between the two set of states. By maximizing this mutual information, the resulting policy learns to better control the environment and can be used to train downstream tasks. The paper experimented with different ways of training the downstream tasks and demonstrated favorable results compared to prior methods.

I think the paper introduces an interesting idea for training unsupervised skills for manipulation tasks. The results also seem very promising. However, I have the following concerns that I hope the authors could help address:

1. The exposition of the paper can be improved. For example, Eq. 4 is referred and discussed multiple times in the text before it is defined. In the first sentence of 3.2, what does adjacent state mean? Does it mean S^s and S^a? What's the difference between \Tau and \Tau' in Eq. 4? In general, although the proposed algorithm seems reasonable, the derivation is a bit confusing.
2. The agent-surrounding separation seems to apply to the full state space, but I'm not sure how we can apply it to observation spaces that contains partial information of the full state. For example, if the robot is equipped with a camera is tasked to train policies with vision input, how does the proposed algorithm handle such cases?
3. I feel that it would be more interesting if the agent-surrounding separation idea could be generalized to be separating the observation space into two subsets where one is optimized to control the other. For example, if we are dealing with a legged robot, which is trained to move forward. The current framework wouldn't be able to handle it because there is no environment state. However, if we separate it into the space of joint angles and velocities, and the space of base position and orientations, would the current framework be able to obtain a policy that can take the robot to a large variety of positions and orientations? If so, a related example would greatly strengthen the paper in my opinion.

---

> ### Author Response · Authors · 2020-11-16
> **To AnonReviewer3**
>
> Thank you for your valuable feedback!
>
> 1. To improve the clarity, we now rearrange the text and first introduce Eq. 4 and then refer to it. We will keep improving the clarity.
> The adjacent state means s_t and s_{t+1}, see Section 3.2 in the third paragraph. The difference between \Tau and \Tau’ is that the trajectory fraction, \Tau’, is defined as the adjacent state pair, \Tau’ = {S_t , S_{t+1}}, and \Tau represents the complete trajectory.
>
> 2. To apply the method to observation spaces that contain partial information of the full state, we can combine our method with state estimation methods, which estimate the state from multiple historical observations. In the case the robot is equipped with a camera, the proposed method can be applied to train policies via maximizing the MI between the agent state and the surrounding state inferred using state estimation methods. We think this is a very interesting direction for future work.
>
> 3. Yes, we agree with you. The idea of the agent-surrounding separation could be generalized to a broader view, such as separating the observation space into two subsets where one is optimized to control the other. Yes, we think that our method could potentially be applied to the legged robot case, where we can treat a subset of the state as the ‘agent’ state and another subset of the state as the ‘surrounding’ state. Then, the robot should learn to control its parts, such as the leg. We think this is another interesting direction worth exploring in the future.

---

> > ### Comment · AnonReviewer3 · 2020-11-24
> > **Thank you for your response**
> >
> > Thanks for your response to my questions. My main concerns have been addressed and I'll increase my score to 7.

---

> > > ### Author Response · Authors · 2020-11-24
> > > **To AnonReviewer3**
> > >
> > > Thank you for your feedback!

---

### Author Response · Authors · 2020-11-16
**Rebuttal Revision**

- We thank the reviewers for their time and valuable feedback!
- We are glad all the reviewers found our paper to be interesting!
- We address the reviewers' comments below and update a revised version of the paper.

---

### Decision · Program_Chairs · 2021-01-07
**Final Decision**

**Decision:**

Accept (Spotlight)

**Comment:**

The paper introduces MUSIC, a method for unsupervised learning of control policies, which partitions state variables into exogenous and endogenous collections and maximizes mutual information between them. Reviewers were uniformly positive, agreeing that the  approach was interesting and well-motivated, and the experiments convincing. Some concerns were raised as to clarity, which were addressed through several revisions of the manuscript. I am happy to recommend acceptance.